# SNP markers revealed the genetic diversity and population structure of *Mesosphaerum suaveolens* (L.) Kuntze Syn. *Hyptis suaveolens* (L.) Poit accessions collected in Benin

Armel Frida Dossa ᴼ*, Dèdéou A. Tchokponhoué ᴼ, Aristide Carlos Houdegbe ᴼ, Enoch G. Achigan-Dako*

Genetics, Biotechnology and Seed Science Unit (GBioS), Laboratory of Plant Production, Physiology and Plant Breeding (PAGEV), School of Plant Sciences, University of Abomey-Calavi, Abomey-Calavi, Republic of Benin

* fridadossa2@gmail.com (AFD); e.adako@gbios-uac.org (EGAD)

## Abstract

*Mesosphaerum suaveolens* (L.) Kuntze is a wild species with many biological activities in medicine. The species can potentially serve as a pesticide in agriculture thanks to its high content of volatile compounds. However, knowledge of the species' genetic diversity and population structure is currently scarce, thus hindering improvement and conservation initiatives. This research used single nucleotide polymorphism (SNP) genotyping to study the diversity and population structure of 175 *Mesosphaerum suaveolens* collected from the three phytogeographical zones of Benin. Our study reports 3,613 high-quality SNP markers for *M. suaveolens.* Based on the markers, a mean polymorphism information content of 0.28, a low expected heterozygosity (He = 0.287) coupled with low genetic differentiation (Fst = 0.007) were observed in the species. The overall average value of observed heterozygosity (*Ho*) was 0.11. The Neighbor-joining analysis identified three subpopulations, whereas two mixed subpopulations were detected via population structure analysis. The results of the molecular variance analysis revealed that differences among samples within populations accounted for the majority of genetic variation (59.3%). The population from the Sudanian region presented the highest genetic diversity (He = 0.293), whereas the Guineo-congolian population presented the lowest genetic diversity (Ho = 0.278). These results pave the way for informed conservation in *M. suaveolens* and highlight the need to expand the genetic base of this species for upcoming improvement initiatives. However, reference genome and genome-wide association studies (GWAS) to associate genomic and morphological or chemical constituents are needed.

**Data availability statement:** All relevant data are within the manuscript and its Supporting Information files.

**Funding:** The author(s) received no specific funding for this work.

**Competing interests:** The authors have declared that no competing interests exist.

## 1. Introduction

Pignut [*Mesosphaerum suaveolens* (L.) Kuntze Syn. *Hyptis suaveolens* (L.) Poit] is a wild species of the Lamiaceae family that originates in the tropical and subtropical climate of America [1]. The species is a heterozygous polyploid with a chromosome number ranging from 2n = 28 to 2n = 32 [2]. This aromatic perennial species is spread worldwide, reaching all continents including Europe, Africa, Asia, and Oceania, and can reach 2 m in height [3,4]. The reproduction of pignut occurs through both self and cross-pollination (geitonogamy and xenogamy). Sexual reproduction facilitated by insect and wind, results in genetic variability within the species [5]. Sharma and Sharma [6] reported that changes from outcrossing to inbreeding contribute to the successful invasiveness of the species. Though considered invasive, *M. suaveolens* is a valuable species due to its essential oils which exhibit high medicinal and insecticidal properties.

Essential oils are environmentally-friendly alternatives to synthetic pesticides, making them a popular choice for pest management globally. The essential oils of *M. suaveolens* exhibit a broad range of biological activities against various storage and field pests thanks to the higher chemical variability of their essential oils. They have been shown to successfully control insects such as *Sitophilus zeamais* Motschulsky (Coleoptera: Curculionidae), *Maruca vitrata* Fabricius (Lepidoptera: Crambidae), *Megalurothrips sjostedti* Trybom (Thysanoptera: Thripidae), and *Aphis craccivora* Koch (Hemiptera: Aphididae) [7–9]. These findings position *M. suaveolens* as a promising aromatic plant for developing novel biological insecticides [10]. In addition to its pest control potential, *M. suaveolens* is considered as an eco-friendly herbicide due to its germination and seedling growth inhibition effect in weeds such as *Cereus jamacaru*. DC. Subsp. *jamacaru* (Cactaceae), and *Echinochloa crus-galli* (L.) P.Beauv. (Poaceae) associated with rice cultivation [11,12]. Beyond its agricultural applications, *M. suaveolens* has several medicinal properties [13]. Its roots, leaves, and twigs are used in decoctions to heal several ailments, including digestive infections, malaria, skin infections, constipation, renal inflammation, headaches, and respiratory issues injuries [8,14]. Additionally, the essential oils exhibit anti-cancer properties against human breast cancer, and possess anti-inflammatory properties [14–17]. Furthermore, the species possesses ecological value owing to its sturdy tolerance to drought and disturbed habitats [18].

Moreover, diversity in the chemical variability of the species essential oils was mentioned in numerous studies. Five *M. suaveolens*-chemotypes have been reported including 1,8-cineole and sabinene, eugenol and germacrene D, fenchone and limonene, sabinene and β-pinene and β-caryophyllene and 1,8-cineole [14,19–22]. In Benin, Salifou et al. [21] identified two of these chemotypes, namely, β-caryophyllene, and 1,8-cineole within all three phytogeographical regions (Guineo–congolian, Sudano–Guinean and Sudanian). This shows the potential for engaging in the species improvement to better exploit its richness.

Despite its potential, previous studies on *M. suaveolens* have focused predominantly on its chemical contents, biological activities against human and crop pathogens [20,23], morphological traits [24], reproductive and pollination ecology [5,6], and

planting modes [25]. While its essential oil application as a biopesticide remains at the laboratory stage, commercialization is hindered by its low essential oil yield. Addressing these challenges requires agromorphological and genetic diversity studies to identify genotypes with higher yield and better oil quality, since variations in *M. suaveolens*' essential oil quality and yield are linked to genotype and environment [10].

To date, information on the genome and genetic diversity structure of *M. suaveolens* is scarce. Meanwhile, other economically significant Lamiaceae species such as *Ocimum gratissimum* [26,27], and *Ocimum basilicum* [28,29], have undergone more detailed genomic studies. The increasing interest in *M. suaveolens*' essential oil for agricultural and medicinal purposes, and lack of information on its genetics, highlight the need for genetic diversity studies to optimize its oil yield and quality.

The key observations that predicted genetic diversity within the species include intrapopulation variations in seed and reproductive organs size [30]. Moreover, two morphological variants of *M. suaveolens*, white-flowering, and blue-flowering, have been reported in India and Brazil [31]. In Benin, empirical observations during fieldwork revealed a third variant with light violet flowers (Fig 1) in the Sudanian phytogeographical zone.

The first molecular evaluation of *M. suaveolens* diversity was conducted using intersimple sequence repeat (ISSR) markers in 2011 [31]. This study with ISSR primers (Hy1- Hy15) analyzed two flower color genotypes (blue and white), revealing distinct genetic basis between them. However, limitations, such as an unweighted degree of difference and limited samples and field observations reduced the study's robustness [31]. Out of 43 polymorphic alleles, 23.6% were unique to the blue flowering form, and 11.1% were exclusive to the white flowering. This was the only one genetic diversity study conducted to evaluate the species' diversity until now. Despite being highly polymorphic, ISSR markers have limitations for resolving geographical differentiation when few genetic markers are used [32]. Hence, it is essential to evaluate the genetic diversity within the species with a high-resolution and robust molecular markers like single nucleotide polymorphism (SNP). SNP assays are an appropriate alternative for diversity analysis in *M. suaveolens* because of their high frequency and availability within the genome, which make them an excellent markers for evaluating genetic diversity [33]. With their saliency, relatively low cost, and accuracy, SNP assays are becoming popular and evolving as first-order markers [34]. Genotyping by sequencing (GBS) facilitates the detection of thousands of SNP markers from many individuals via a reduced representation library [35]. One of the most affordable, simple, and effective genotyping-by-sequencing platforms is the Diversity Array Technology Sequencing (DArTseq) platform, which enables easy identification of genome-wide markers via restriction enzymes, as well as sequencing of restriction elements [36].

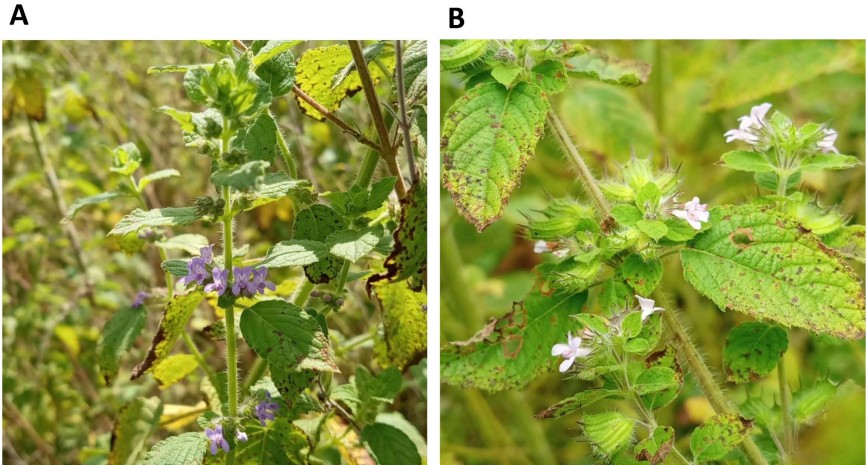

**Fig 1. Inflorescence color-defined *Mesosphaerum suaveolens* morphotypes in Benin.** (A): morphotype with a purple inflorescence; (B): morphotype with a light-purple inflorescence.

DArTseq technology generates a large pool of high-quality SNPs and SilicoDArT markers, even in species without reference genome. Furthermore, DArTseq is an excellent approach for genotyping a wide number of individuals [34,37,38]. It has been used to reveal the genetic diversity and population structure of 62 accessions of aromatic basil (*Ocimum basilicum* L.) from Lamiaceae family in Ethiopia [38]. Numerous molecular studies have used DArTseq to explore the genetic diversity, to identify markers of interest, and assess the relationship among species lines of several crops such as *Vigna unguiculata* (L) Walp. [37], *Colocasia esculenta* (L.) Schott [39], *Synsepalum dulcificum* (Schumah & Thonn.) Daniell [40], *Zea mays* L. [41], *Phaseolus vulgaris* L. [42], *Musa acuminata* Colla [43] and *Macrotyloma geocarpum* (Harms) Maréchal & Baudet [44].

Despite the widespread use of single nucleotide polymorphisms in genetic diversity studies, no research has yet reported the genetic diversity of *M. suaveolens* using SNP markers worldwide. This represents a significant gap that our study seeks to address. By generating SNP markers from *M. suaveolens*, we will make this species' first sequences accessible and lay a good foundation for further application in studying the species' evolutionary history, conservation, genetic mechanism regulating essential oil yield and quality, and breeding purposes. The aim of this research was to generate molecular information of *M. suaveolens* that will facilitate genetic improvement of *M. suaveolens* accessions for improved essential oil yield and quality. Specifically, we aimed to assess the extent of genetic diversity and population structure in *M. suaveolens* germplasm from Benin using high-density SNP markers. To achieve this, the study seeks to answer the following research questions: 1) what is the extent of genetic diversity in the *M. suaveolens* germplasm studied? 2) does geographical distribution influence the genetic diversity of a species? 3) what is the genetic structure of the collected accessions? We hypothesized that 1) the accessions of *M. suaveolens* from close geographical origins presented molecular similarities and 2) the *M. suaveolens* collection encompassed more than one genetic cluster.

## 2. Materials and methods

### 2.1. Leaf sampling strategy

A total of one hundred seventy-five pignut accessions were sampled and georeferenced (S1 Table) with the approval of the local authorities. The collection covered three populations corresponding to the phytogeographical regions of Benin (the Guineo-congolian, Sudano-guinean, and Sudanian regions) from December 2019 to March 2020, with at least a minimal distance of 10 m between two accessions.

The Guineo-congolian zone, located in the southern part of Benin, has a subequatorial climate. It is characterized by a bimodal season, with two rainy seasons and two dry seasons. The annual precipitation ranges from 900 to 1,400 mm. The majority of soils are ferralitic, clayey, sandy, hydromorphic, low-lying, and vertisols. The vegetation of this zone comprises a mosaic of patches of dense rainforest, mangrove swamps, grasslands, savannahs, and fallow land [45]. The Sudano-guinean zone is the transition region. It is characterized by a unimodal season with an average rainfall ranging from 900 to 1,110 mm. The mineral soils here are not particularly well-developed or fertile, and the ferruginous soils on crystalline bedrock are variable in terms of fertility. The vegetation consists of wooded and shrubby savannahs, open forests, dense dry forests, and forest galleries. The Sudanian zone is distinguished by unimodal rainfall regime. The average annual precipitation ranges from 900 to 1,100 mm. The soils include hydromorphic, ferrallitic, lithosols, and drained soils (61,63).

A total of 93 accessions were sampled from the Guineo-congolian population, 46 from the Sudano-guinean population, and 36 from the Sudanian population. For each accession, 10 pest-free young leaves were sampled and automatically silica gel-dried [40] before being transported to the laboratory [Genetics, Biotechnology and Seed Science Unit (GBioS) in Abomey-Calavi]. The labelled accessions were then arranged in kits of 96 tubes each and shipped to SEQART AFRICA at the International Livestock Research Institute (ILRI) for genotyping.

## 2.2. National and local approval

Before embarking on the fieldwork to collect accessions, the first author secured the approval of the academic committee of the École Doctorale des Sciences Agronomiques et de l'Eau (EDSAE) of the Faculty of Agricultural Sciences (FSA) of the University of Abomey-Calavi (UAC). The ethics committee that waived the study was the first author's thesis committee at the University of Abomey-Calavi, comprising Prof. Enoch G. Achigan-Dako, Prof. Eleonore Yayi-Ladekan, Prof. Bokonon-Ganta Aimee, Prof. Latifou Lagnikan, and Prof. Victorien T. Dougnon. The collection of the plant materials adhered to both the National and International Code of Conduct for Plant Germplasm collection. Prior to collecting samples, the objectives of the research project were presented to the local authorities of each village under investigation. The informed verbal consent of all local authorities was obtained before sample collection.

## 2.3. DNA extraction and sequencing

The extraction of DNA was performed with a Nucleomag Tissue Kit (DArtseq proprietary kit). The amount of isolated genomic DNA ranged between 50 and 100 ng/µl. On 0.8% agarose, the quantity and purity of the DNA were examined [42,44]. Libraries were constructed following Kilian et al. [46]. The genome complexity reduction approach used was that of Diversity Arrays Technology and Sequencing (DArTseq), whereby PstI and MseI enzymes were combined to digest the genomic DNA and then to ligate barcoded and common adapters [47]. The constructed libraries were sequenced via 77-base single-read sequencing runs. Next-generation sequencing was performed via HiSeq 2500, and DArTsoft14 was used to score the DArTseq SNP markers. The markers were scored as 0 (reference homozygous), 1 (homozygous), or 2 (heterozygous). The genomic matrix obtained was preliminarily quality-assessed with the "dartR package" [48] in the R environment (version 4.1.2). Only SNPs with a call rate >85%, a minor allele frequency (MAF) > 0.05, and a reproducibility rate >95% were used [40,49] to constitute the genomic matrix of the 175 *M. suaveolens* accessions. Missing values of that genomic matrix were imputed on the KD computational platform (https://kdcompute.seqart.net/kdcompute/plugins) via the Ensemble Method algorithm to obtain a full genomic matrix for the analysis described below.

## 2.4. Genetic diversity analysis

First, the quality of the retained markers was explored through analysis of their polymorphic information content (PIC). The following overall genetic diversity indices were subsequently computed: expected heterozygosity (*He*), observed heterozygosity (*Ho*), total gene diversity (*Ht*), and the inbreeding coefficient ($F_{IS}$). They were all estimated with the "dartR package" [48] in R (version 4.1.0). To determine the genetic diversity among populations, *He*, *Ho*, allelic richness, and $F_{IS}$ were calculated across populations. To ascertain the existence of private alleles, the "poppr package" [50] was used. The overall differentiation index (Fstg) was calculated to assess the genetic differentiation within the studied germplasm, whereas pairwise genetic differentiation indices (Fstp) were computed between pairs of populations to understand the level of divergence between them [51].

## 2.5. Population structure

The population structure was assessed via three methods. First, an analysis of molecular variance was performed in the R environment (version 4.1.0) to estimate the relative contribution of variation sources (among populations, among samples within populations, within samples) to genetic variation in the species [39]. Second, STRUCTURE software version 2.3.4 [52] was used to implement Bayesian model clustering. Hypothetical number of clusters (K) ranged from 2 to 10, with an initial burn-in period of 10,000 for each run of 10,000 Markov Chain Monte Carlo (MCMC) iterations. The ΔK method was used to determine the highest predicted value of clusters (K) for each test via the online program Structure Harvester [53,54]. The average silhouette method was used at the appropriate K-value to determine the distribution of likelihood values of K, which exhibits a notable inflection point that was discernible in the Delta K-values. Only accessions with a

membership coefficient ≥ 0.7 for a cluster were classified as belonging to that cluster [55]. Consequently, accessions with a membership coefficient < 0.70 at every given K were categorized under the admixture group. Lastly, the relationships among the 175 *M. suaveolens*, a phylogenetic tree, were assessed via neighbor-joining tree analysis via Molecular Evolutionary Genetics Analysis X (MEGAX) software [56] based on the maximum composite likelihood evolutionary distance. Visualization was performed via the interactive tree of life (ITOL, V.5) [57] web application (https://itol.embl.de/).

## 3. Results

### 3.1. Markers quality

Among the 18,205 raw SNP markers generated, 3,613 were ultimately retained after the quality assessment. These 3,613 markers have average polymorphism information content (PIC) values ranging from 0.09 to 0.49, with a mean value of 0.28 ± 0.002. The mutations noted in the DNA sequences are transitions (59.5%) and transversions (40.5%), with a transition/transversion rate of 1.5%. The proportions of C/T and A/G transitions were 31.4% and 28.1%, respectively. The proportions of A/T, A/C, G/T, and G/C transversions were 11.4%, 10.4%, 9.7%, and 8.4%, respectively. Minor allele frequency had an average value of 0.2 ± 0.1 and varied from 0.05 to 0.5.

### 3.2. Genetic diversity in *M. suaveolens* populations

The overall average value of observed heterozygosity ($Ho$) was 0.117, whereas the total gene diversity ($Ht$), expected heterozygosity ($He$), inbreeding coefficient ($F_{IS}$), and interpopulation diversity ($D_{ST}$) in the species were 0.289, 0.287, 0.592, and 0.002, respectively. The estimates of these statistics for each studied population are presented in Table 1. While expected heterozygosity estimates increased following the south-north ecological gradient, neither the observed nor the allelic richness followed this trend. Although the mean allelic richness seemed not statistically different among population, the Guinean population presented the highest value, whereas the Sudano-guinean population presented the lowest value for this parameter. For the observed heterozygosity, the Sudano-guinean population had the highest estimate (0.135), with the lowest level of inbreeding.

No private alleles were detected among the three populations. However, our analysis identified one allele that was specific to the Sudanian and Sudano-guinean populations when compared to the Guineo-congolian population. The set of Guineo-congolian and Sudano-guinean populations shared 27 private alleles by comparing them with the Sudanian population. Similarly, 20 alleles were specific to the Guineo-congolian and Sudano-guinean populations when compared with the Sudanian population (Table 2).

### 3.3. Genetic differentiation

The overall genetic fixation index (Fstg) value for *Mesosphaerum suaveolens* was quasi-null (Fstg = 0.007), suggesting very low differentiation trend in the species. The pairwise differentiation indices between the pairs of studied populations

**Table 1. Diversity statistics for the 175 *Mesosphaerum suaveolens* using 3,613 DArTseq-based single nucleotide polymorphism (SNP) data sets.**

| Population | $Ho$.adj | $uHe$ | $F_{IS}$ | Allelic richness |
|---|---|---|---|---|
| Guineo-congolian | 0.104 | 0.278 | 0.625 | 1.991 |
| Sudano-guinean | 0.135 | 0.284 | 0.523 | 1.985 |
| Sudanian | 0.111 | 0.293 | 0.618 | 1.986 |
| **Overall average values** | **0.117** | **0.287** | **0.592** | |

$Ho$.adj, observed heterozygosity; $He$, expected heterozygosity; $F_{IS}$, inbreeding coefficient.

are shown in Table 3. The greatest pairwise differentiation index (Fstp = 0.018) was observed between the Sudanian and Guineo-congolian populations, whereas the lowest (Fstp = 0.006) was recorded between the Sudano-guinean and Sudanian populations (Table 3).

### 3.4. Population structure

The results of the hierarchical analysis of molecular variance (AMOVA) revealed that the genetic variation observed was mainly due to variation among samples within populations (59.3%), whereas variation within samples accounted for 39.6% of the total genetic variation (Table 4). Only 1.1% of the molecular variation was due to variation among populations (phytogeographical regions).

The population structure inferred from the Bayesian clustering approach suggested an optimal value of K = 2 (S1 Fig), thus defining two clusters for the 175 accessions (Fig 2). Based on membership probabilities of ≥ 0.70 (S2 Table), cluster 1 (C1) comprises 18 accessions, and cluster 2 (C2) comprises 140 accessions. The remaining 17 accessions were placed in the admixture group. Cluster 1 (C1) encompassed individuals from all regions, with 5.5% belonging to Sudanian, 5.5% belonging to Sudano-guinean, and 89% belonging to Guineo-congolian regions. Cluster 2 (C2) comprised 48.6% Guineo-congolian accessions. Guineo-congolian accessions (58.8%) also largely dominated the admixture group.

The accessions were structured based on the neighbor-joining tree, and the 175 accessions were classified into three groups: group I, group II, and group III. Group I, consists of 14 accessions. Group II includes 25 accessions, and group

**Table 2. Alleles privacy among groups of sampled populations.**

| Comparison between Pop1 and Pop2 | | Priv1 | Priv2 | Total priv | AFD |
|---|---|---|---|---|---|
| Pop1 | Pop2 | | | | |
| Guineo-congolian (GC) | S+SG | 0 | 1 | 1 | 0.059 |
| Sudano-guinean (SG) | GC+S | 0 | 20 | 20 | 0.050 |
| Sudanian (S) | GC+SG | 0 | 27 | 27 | 0.068 |

GC, Guineo-congolian; SG, Sudano-guinean; S, Sudanian; Pop1, reference population; Pop2, All other population except the reference population; AFD, Allelic fixed difference; Priv 1, Reference population's Private allele; Priv 2, Private allele of all other population except the reference population.

**Table 3. Pairwise Cockerman genetic fixation index (Fst) values between _Mesosphaerum suaveolens_ populations.**

| Populations | Guineo-congolian | Sudano-guinean |
|---|---|---|
| **Sudano-guinean** | 0.007 | |
| **Sudanian** | 0.018 | 0.006 |

**Table 4. Analysis of molecular variance (AMOVA) based on three phytogeographical regions (Guineo-congolian, Sudano-guinean, and Sudanian), 175 individual accessions, and 3,613 single nucleotide polymorphism (SNP) markers.**

| Source of variation | df | Sum of Squares | Mean Squares | % Variation | P-value |
|---|---|---|---|---|---|
| Among populations | 2 | 2866.77 | 1433.38 | 1.1 | < 0.001 |
| Among samples within populations | 172 | 141573.92 | 823.10 | 59.38 | < 0.001 |
| Within samples | 175 | 36072.5 | 206.12 | 39.6 | < 0.001 |
| Total | 349 | 180513.19 | 517.22 | 100.0 | |

df = degrees of freedom.

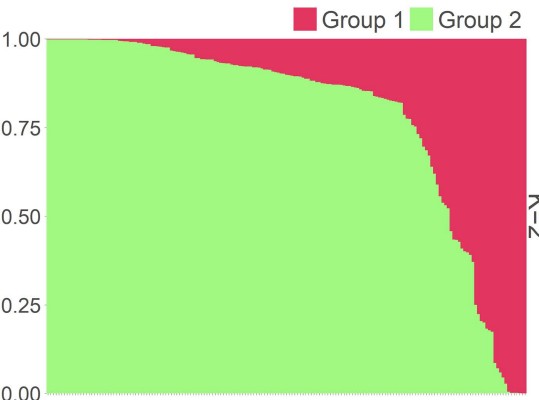

**Fig 2. Barplot showing the structuring pattern of the 175 *Mesosphaerum suaveolens* accessions.**

III comprises 136 accessions (77.8%). The largest group III includes some individuals from cluster 2 (C2), along with all accessions in cluster 1 (C1) identified by STRUCTURE, as well as all admixed accessions (Fig 3).

## 4. Discussion

The present study is the first to depict the extent of genetic diversity in pignut (*M. suaveolens*) using DArTseq technology with 3,613 high-quality SNP markers. This generates basic molecular information about *M. suaveolens* for future research linked to conservation and breeding. Among the identified SNP variants, A/G and C/T were the most common SNP variants in the studied *M. suaveolens* accessions. This finding is consistent with previous reports in both wild (*Amorphophallus paeoniifolius*) [58] and cultivated species (*Vigna unguiculata* and *Musa acuminata* cv. Sotoumon) [37,43,55], where transitions were the common source of genetic variation. In this study, the retained SNPs were moderately informative, as the Polymorphism Information Content (PIC) values fell within the range of 0.25–0.5 [59]. This informativeness is similar to values reported in an Ethiopian *Ocimum basilicum* study, which varied from 0.23 to 0.37 using DArTseq [38]. Overall, the markers used were informative enough to help appreciate the extent of diversity in the species. The moderate PIC value of the SNP markers in this study may be enhanced by a large number of markers and a wide genome coverage [42].

The average expected heterozygosity (*He*) was 0.287, indicating low genetic diversity in the studied *M. suaveolens* germplasm. This result corroborates the findings of Gossa et al. [38], where *O. basilicum* populations exhibited low diversity (*He* varied from 0.13 to 0.33) when evaluated using SNP markers. Similarly, Kyrkjeeide et al. [60] reported comparable expected heterozygosity values (*He* = 0.27) for *Dracocephalum ruyschiana* accessions, an aromatic plant from the Lamiaceae family [60]. Low to moderate diversity was also observed in *Hyptis pectinata*, a close relative of *M. suaveolens* in Brazil using ISSR markers [59]. Given that widely distributed wild species such as *M. suaveolens* typically exhibit a relatively high level of diversity, the unexpected low genetic diversity observed in this study may be attributed to various factors. First, the geographical coverage of the study, although encompassing three demarcated ecological regions of Benin, is still limited to the Dahomey Gap, known as a single ecological area in the African rainforest space. We then speculated that expanding the collection areas to other countries or African ecological regions could potentially increase the diversity level [61]. Therefore, efforts must be undertaken to collect and preserve the genetic diversity of *M. suaveolens* present in Benin and worldwide. Second, the species was previously reported to exhibit both autogamy and allogamy, and the low diversity observed might indicate the preponderance of autogamy in the species, as was experimentally confirmed by Aluri and Reddi [62]. In addition, *M. suaveolens* species is facing landscape fragmentation in Benin, a phenomenon known to

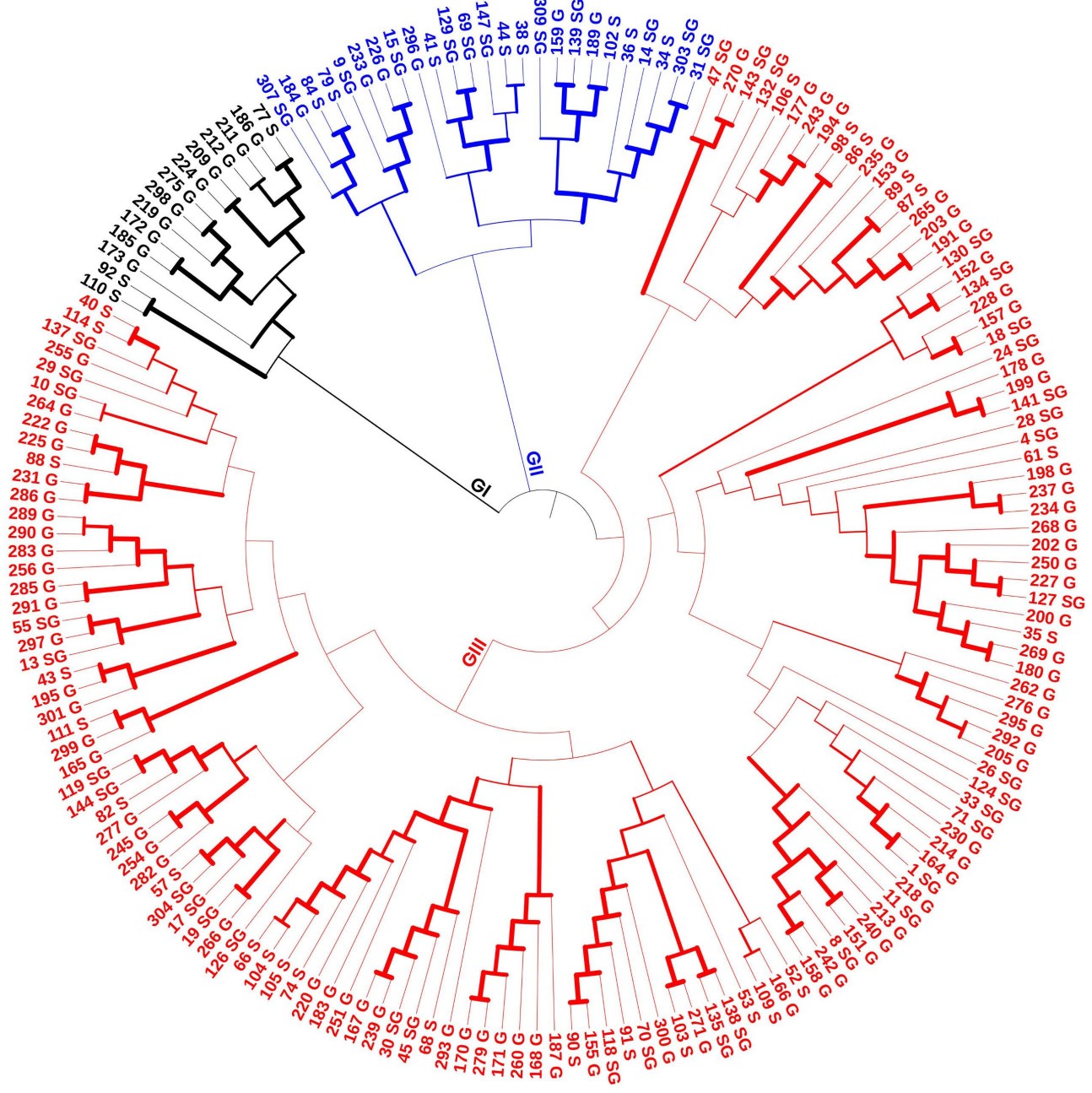

**Fig 3. Neighbor-joining tree depicting the relationship among the 175 *Mesosphaerum suaveolens* accessions based on 3,613 single nucleotide polymorphism (SNP) markers.** Group I (Color black), Group II (Colour Blue), and Group III (Color red).

reduce the effective size of wild populations and, in turn limits genetic variation [58]. Given this phenomenon, the assessment of genetic diversity is fundamental to select worthwhile genotypes for future biological research studies. Furthermore, the populations' size involved in this study could explain the low diversity observed, as it is reported that genetic diversity increases significantly as population size grows [63].

The low diversity in the present study, coupled with a moderate inbreeding coefficient ($F_{IS}$ = 0.592), indicates a moderate level of fixation. This value of inbreeding could also justify the presence of both autogamy and allogamy in the species [5]. Among the studied populations, the Sudanian population exhibited the highest diversity, suggesting that it may have more potential to adapt to the changing climate in a region already known to experience severe climatic conditions. Therefore, this population is a potential source of genes of interest that could be deployed in improvement programs to develop specific product profile genotypes. Hence, there is an urgent need for more concentrated actions to collect genetic resources throughout the region for ex-situ conservation strategy profiling. Using SNP markers allowed an evaluation of the relationships for new future breeding or conservation efforts. The fixation index (Fst) measures a population's differentiation linked to genetic structure [64]. An Fst value > 0.25 indicates very high differentiation, and an Fst < 0.05 implies low differentiation [65]. The values ranging from 0.05 to 0.15 and 0.15 to 0.25 reflected moderate and high differentiation, respectively. The Fst value (Fstg = 0.007) observed in this study indicates very low genetic differentiation between these three populations. Similarly, a low genetic differentiation among populations was reported for many other species from the Lamiaceae family, including *Dracocephalum ruyschiana* and *Dracocephalum austriacum* [66]. The very low differentiation observed in this study may also be associated with regions of samples collection or distance among the studied populations. As shown in Table 3, the closer the populations are, the smaller their genetic distances are, and the further apart they are, the greater their genetic distances are. The Fst value also indicates high gene flow among populations. Gene flow, which enhances genetic diversity within populations and reduces diversity among populations, is a factor affecting the genetic differentiation of populations, contrary to the effect of genetic drift [67]. Yao et al. [68] observed low differentiation in cowpea (*Vigna unguiculata*) populations due to long-distance gene dispersal, either by pollen or by seed. This mechanism could also explain the low differentiation noted in this study, as the sexual reproduction in *M. suaveolens* is possible through wind and insects contribution [62]. However, the current very low differentiation between *M. suaveolens* populations is contrasting the findings of Salifou et al. [21] based on the chemical constituents of the essential oil. Salifou et al. [21] reported two chemotypes of *M. suaveolens* according to the three phytogeographical regions of Benin, β-caryophyllene (Sudanian and Sudano-guinean), and 1,8-cineole (Guineo-congolian), which is somewhat predictive of genotype grouping based on accessions' location. The accumulation of bioactive compounds in *M. suaveolens* varies across different regions due to environmental factors, genetic diversity, elicitors, and ecological conditions [17,69,70]. Studies highlighted that elements like soil composition, climate, and exposure to biotic and abiotic stress influence the production of secondary metabolites in this species [71,72]. The low diversity coupled with very low differentiation observed in this study suggests that, beyond genetic factors, environmental factors are key elements to be considered in trait enhancement, particularly in the improvement of the essential oils, which is the primary focus for *M. suaveolens*. This finding further highlights the need to expand the collection of accessions to access greater genetic variability. Selecting genotypes that remain stable across different environments in terms of yield and essential oil composition is therefore vital, reinforcing the importance of integrating genetic data with the chemical profiling of essential oils.

Differences in population structure display genetic variation and highlight a species' capability to adapt to a changing habitat [73]. The neighbor-joining analysis detected three clusters, whereas the STRUCTURE analysis revealed two genetic groups. This difference can be partly explained by the restriction of the software STRUCTURE to accurately handle population structuring [44]. In addition to this limitation, the results of STRUCTURE are influenced by factors such as markers type, the number of studied populations, sample size, and number of loci scored [54]. Similar mismatching between neighbor-joining and structure analysis have been reported by Akohoué et al. [44] and Tchokponhoué et al. [40] while structuring Kersting's groundnut and the Sisrè berry populations, respectively. Moreover, the Bayesian STRUCTURE analysis identified two groups, and 9.7% of the 175 *M. suaveolens* accessions were admixed. Pyne et al. [74] reported a 15.6% admixture among 180 *Ocimum* spp. accessions studied. This level of admixture could further support the assumption of genomic content exchange, possibly during cross-pollination [74]. This phenomenon has also been observed in *Elsholtzia stauntonii* Benth. (Lamiaceae) [61]. The results of this study indicate that

natural outcrossing has preserved genetic diversity within *M. suaveolens* accessions. Furthermore, some accessions collected in the same districts were structured into different clusters, indicating that natural selection, genetic mutation, and migration could be the main mechanisms of evolution that affect these genetic differences [75]. According to the neighbor-joining tree, 77.8% of the sampled accessions were grouped into a single cluster which included both blue flowering and light purple flowering *M. suaveolens*). The accessions in this group were collected from diverse habitats, including mountainous region, easy-to-access areas, and abandoned fields [76], where anthropic activities and animals are permanent. The species' habitat was mostly found along roadside. *M. suaveolens* produces abundant, lightweight, and small seeds that can be easily transported without difficulty by water and wind over long distances. This could explain the high gene flow observed among populations and accessions. The genetic similarity among accessions further supports this hypothesis.

To fully understand the population divergence of *M. suaveolens*, extensive germplasm collection across different regions of the world is necessary. Considering the low genetic diversity within *M. suaveolens* and the very low differentiation among populations, broadening the genetic base within the species could be helpful for future improvement programs. This will require cross-pollination. Cross-pollination increases genetic diversity through introducing new alleles and increasing variation [63]. In the current case, cross-pollination could enhance the genetic diversity by crossing the studied populations with another genetically distant population. Thus, introducing new germplasms and facilitating crosses between genotypes from different continents and countries would be essential steps toward to increase the genetic diversity of *M. suaveolens*.

## 5. Conclusion

This SNP-based characterization study generated the first wide genetic polymorphism information in *M. suaveolens*. A total of 3,613 SNP markers with minor allele frequency ≥ 0.05 were discovered. A low genetic diversity within *M. suaveolens* accessions and a very low differentiation among studied populations were observed. Two distinct genetic clusters were identified in the collected accessions; yet the genetic diversity was not structured according to phytogeographical regions. Among the studied populations, the Sudanian population exhibited higher diversity than the Guineo-congolian and Guineo-sudanian populations. The discovered DArTseq SNP markers represent a novel genomic resources to improve *M. suaveolens* population management and breeding program. These resources could aid in improving traits like essential oil yield and quality, and the biosynthesis of bioactive compounds. The findings open rooms for the genome assembly of *M. suaveolens* and further research combining morphological, chemical constituents, and molecular analysis to allow genome-wide association studies. Worldwide, this study is the first to describe the population structure and genetic diversity in *M. suaveolens* using SNP, thus setting the scene for a proper marker-assisted improvement of useful traits in the species.

## Supporting information

**S1 Table. Coordinates for each accession from the three phytogeographical regions of Benin.** This is a word file presenting the coordinates of each accession sampled.
(DOCX)

**S2 Table. *Mesosphaerum suaveolens* accessions grouping based of the membership value (Q).** Admixt = Admixed individuals, C1 = Individuals of the group 1 and C2 = Individuals of the group 2. This is a word file presenting the accessions grouping.
(DOCX)

**S1 Fig. Estimate of the optimal number of clusters with K ranging from 1 to 10.**
(PDF)

## Acknowledgments

The first author wishes to thank the West African Research Association (WARA) for its support in accessions collection. Furthermore, the first author would like to express gratitude to the Genetics, Biotechnology and Seed Science Unit (GBioS) of the University of Abomey-Calavi for overall assistance.

## Author contributions

**Conceptualization:** Armel Frida Dossa, Enoch G. Achigan-Dako.

**Data curation:** Dèdéou A. Tchokponhoué.

**Formal analysis:** Dèdéou A. Tchokponhoué.

**Funding acquisition:** Armel Frida Dossa, Enoch G. Achigan-Dako.

**Investigation:** Armel Frida Dossa.

**Methodology:** Armel Frida Dossa, Enoch G. Achigan-Dako.

**Project administration:** Armel Frida Dossa, Enoch G. Achigan-Dako.

**Software:** Dèdéou A. Tchokponhoué.

**Supervision:** Enoch G. Achigan-Dako.

**Validation:** Dèdéou A. Tchokponhoué, Enoch G. Achigan-Dako.

**Visualization:** Dèdéou A. Tchokponhoué.

**Writing – original draft:** Armel Frida Dossa.

**Writing – review & editing:** Dèdéou A. Tchokponhoué, Aristide Carlos Houdegbe, Enoch G. Achigan-Dako.

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
