## [Decision Letter · Decision Letter 0]

17 Oct 2024

Dear Dr. Dossa,

Thank you for submitting your manuscript to PLOS ONE. After careful consideration, we feel that it has merit but does not fully meet PLOS ONE’s publication criteria as it currently stands. Therefore, we invite you to submit a revised version of the manuscript that addresses the points raised during the review process.

**ACADEMIC EDITOR:**

Dear Dr. Dossa, thank you for submitting your manuscript for consideration in Plos One. I have received the reviewers' comments on your work.

Please review your manuscript point by point and provide an updated version of your work.

We look forward to receiving your revised manuscript.

Kind regards,

Mozaniel Santana de Oliveira, Ph.D

Academic Editor

PLOS ONE

Journal Requirements:

4. In your Methods section, please provide additional information regarding the permits you obtained for the work. Please ensure you have included the full name of the authority that approved the field site access and, if no permits were required, a brief statement explaining why.

6. We note that Figure 1 in your submission contain copyrighted images. All PLOS content is published under the Creative Commons Attribution License (CC BY 4.0), which means that the manuscript, images, and Supporting Information files will be freely available online, and any third party is permitted to access, download, copy, distribute, and use these materials in any way, even commercially, with proper attribution. For more information, see our copyright guidelines: http://journals.plos.org/plosone/s/licenses-and-copyright.

7. We note that Figure 1 in your submission contain map/satellite image which may be copyrighted. All PLOS content is published under the Creative Commons Attribution License (CC BY 4.0), which means that the manuscript, images, and Supporting Information files will be freely available online, and any third party is permitted to access, download, copy, distribute, and use these materials in any way, even commercially, with proper attribution. For these reasons, we cannot publish previously copyrighted maps or satellite images created using proprietary data, such as Google software (Google Maps, Street View, and Earth). For more information, see our copyright guidelines: http://journals.plos.org/plosone/s/licenses-and-copyright.

Reviewers' comments:

Reviewer's Responses to Questions

**Comments to the Author**

1. Is the manuscript technically sound, and do the data support the conclusions?

Reviewer #1: Yes

Reviewer #2: Partly

2. Has the statistical analysis been performed appropriately and rigorously?

Reviewer #1: Yes

Reviewer #2: No

3. Have the authors made all data underlying the findings in their manuscript fully available?

Reviewer #1: Yes

Reviewer #2: No

4. Is the manuscript presented in an intelligible fashion and written in standard English?

Reviewer #1: Yes

Reviewer #2: No

Reviewer #1: This work would be more complete if you have incorporated data on those important traits you mentioned in the manuscript. After all, genetic diversity is relevant to breeders and conservationists. For breeder the molecular diversity has to be supported by morphological data where he can see if the diversity is functional and because of certain traits (past artificial or natural selection). Otherwise if the diversity is mainly because of SNPs that have no agronomic importance, it will have no use for the breeder. Hence I strongly recommend to incorporate data on a few traits and see after doing PCA or NJ-tree to see if those traits are determinants of population structure and differentiation.

Reviewer #2: Dears …

The genetic diversity of Mesosphaerum Suaveolens was examined by the researchers using molecular markers. There is scant fresh information pertaining to the subject of this investigation. Further calculations of parameters are required. The following adjustments are necessary:

Abstract

More scored data should be inserted

The researchers should be performed the unique bands and diversity index

The sentences regarding the conclusion and future prospects should be incorporated by the authors at the conclusion of the abstract.

Keywords

The terminology in the title should not be utilized as keywords. This requires a change to the keyword framework.

Introduction

It is better to use the recent references

Detailed information regarding the genetic diversity and its varieties should be included.

Detailed information regarding the genetic diversity methods should be included

It would be beneficial to include a few lines regarding the plant species' beneficial applications.

Why did the researchers select this plant species?

More information about the used markers should be stated

The hypothesis is inaccurately articulated. The authors ought to incorporate a brief section delineating the knowledge gaps they sought to address in their research.

Furthermore, authors must provide a statement emphasizing the uniqueness of their findings in the conclusion.

What novel research-related connections or activities have the authors identified in this study that were absent in prior studies?

It is imperative to address both the general and specific objectives.

Materials and Methods

A map showing the samples collection should be included

The voucher number of this plant species should be provided

It is necessary to provide the entire name of each acronym.

References must be provided to substantiate each technique.

The researchers should be performed the unique bands, genetic and diversity index for each marker and population

The manufacture name of all materials should be added

The data analysis section should be located at the end of MM

Results and discussion

All captions should be improved.

It is necessary to include definitions for all abbreviations in the captions of tables and figures.

Creating a PCA plot should be provided

The dendrogram should be annotated with the number of clusters.

What are the criteria used to determine the number of clusters?

The discourse lacked conviction. The commentary is insufficient and necessitates updating, as most phrases reiterate the findings instead of offering a critical examination of the data. The authors must elucidate the importance of the findings from each study in connection to their own research. The authors are accountable for elucidating the findings about the diversity index, clustering, PIC, and He. The authors ought to elucidate the disparity in the accumulation of bioactive across various areas.

Conclusion

The researchers ought to provide a concise summary of the most significant findings, as this section is presented in an easily understandable format. Future research should incorporate additional investigation on this subject.

Best regards

**Do you want your identity to be public for this peer review?** For information about this choice, including consent withdrawal, please see our Privacy Policy

Reviewer #1: **Yes: ** Wosene G. Abtew

Reviewer #2: **Yes: ** Nawroz Tahir

---

## [Author Response · Author response to Decision Letter 1]

17 Mar 2025

The response to reviewers comments has been attached as a word file named "Response to reviewers.docx".

---

## [Decision Letter · Decision Letter 1]

13 Apr 2025

Dear Dr. DOSSA,

Thank you for submitting your manuscript to PLOS ONE. After careful consideration, we feel that it has merit but does not fully meet PLOS ONE’s publication criteria as it currently stands. Therefore, we invite you to submit a revised version of the manuscript that addresses the points raised during the review process.

**ACADEMIC EDITOR:**

Dear Dr. Dossa, I have received the reviewers' comments on your manuscript and have made a decision.

I suggest major revision, carefully review each issue raised by the reviewers and return with a revised version of your manuscript.

We look forward to receiving your revised manuscript.

Kind regards,

Mozaniel Santana de Oliveira, Ph.D

Academic Editor

PLOS ONE

Reviewers' comments:

Reviewer's Responses to Questions

**Comments to the Author**

Reviewer #2: (No Response)

Reviewer #3: (No Response)

2. Is the manuscript technically sound, and do the data support the conclusions?

Reviewer #2: Partly

Reviewer #3: Yes

3. Has the statistical analysis been performed appropriately and rigorously?

Reviewer #2: Yes

Reviewer #3: Yes

4. Have the authors made all data underlying the findings in their manuscript fully available?

Reviewer #2: No

Reviewer #3: No

5. Is the manuscript presented in an intelligible fashion and written in standard English?

Reviewer #2: Yes

Reviewer #3: No

Reviewer #2: Dear Authors

The following points should be modified

1. too many references under 2020 are used in this manuscript. Please minimize the number of references and the authors should use the recent references

2. The conclusion is too long and should be summarized

3. The discussion needs an improvement

Reviewer #3: The results will inform the breeding and conservation of the species, so it has scientific merit. The data analysis is also okay. However, the introduction needs to be re-written for clarity and flow. The entire manuscript needs to be re-read and edited to minimize grammatical errors.

**Do you want your identity to be public for this peer review?** For information about this choice, including consent withdrawal, please see our Privacy Policy

Reviewer #2: **Yes: ** Nawroz Tahir

Reviewer #3: No

---

## [Author Response · Author response to Decision Letter 2]

15 Jun 2025

The response to specific reviewer and editor comments has been attached as a word file named "Response to reviewers.docx".

---

## [Decision Letter · Decision Letter 2]

20 Jul 2025

Dear Dr. DOSSA,

Thank you for submitting your manuscript to PLOS ONE. After careful consideration, we feel that it has merit but does not fully meet PLOS ONE’s publication criteria as it currently stands. Therefore, we invite you to submit a revised version of the manuscript that addresses the points raised during the review process.

We look forward to receiving your revised manuscript.

Kind regards,

Mozaniel Santana de Oliveira, Ph.D

Academic Editor

PLOS ONE

Journal Requirements:

Additional Editor Comments (if provided):

Dear authors, I have received the necessary revisions of your manuscript, please review each question point by point and send a revised version of your manuscript.

Best regards

Mozaniel Santana de Oliveira Ph.D.

Academic Editor

Reviewers' comments:

Reviewer's Responses to Questions

**Comments to the Author**

Reviewer #3: (No Response)

2. Is the manuscript technically sound, and do the data support the conclusions?

Reviewer #3: Yes

3. Has the statistical analysis been performed appropriately and rigorously?

Reviewer #3: Yes

4. Have the authors made all data underlying the findings in their manuscript fully available?

Reviewer #3: Yes

5. Is the manuscript presented in an intelligible fashion and written in standard English?

Reviewer #3: Yes

Reviewer #3: The manuscript has greatly improved. All the issues raised previously have been attended to. A few minor comments have been added.

**Do you want your identity to be public for this peer review?** For information about this choice, including consent withdrawal, please see our Privacy Policy

Reviewer #3: No

---

## [Author Response · Author response to Decision Letter 3]

28 Jul 2025

Responses to reviewers' comments

We thank the editor and the reviewer for their comments. We provided below a point-by-point response to their comments.

Reviewer #3

Comment #3.1. For consistence, I suggest you use both ‘improvement and conservation’ on L18 (mentions only improvement) since L31-32 use both terms.

Response #3.1. We thank the reviewer for this recommendation. Conservation was added to improvement (See line 18 of the revised manuscript with track changes).

Comment #3.2. L40-41 revise to ‘subtropical climate of America’

Response #3.2. The correction was made as recommended. Please see lines 40-41 of the revised manuscript with track changes.

Comment #3.3. L42- ‘perennial’ is more commonly used than pluriannual. You many consider to change

Response #3.3. Pluriannual was replaced by perennial. (See line 42 of the revised manuscript with track changes)

Comment #3.4. L46-Need to change the order the words from ‘xenogamy and geitonogamy’ to ‘respectively geitonogamy and xenogamy’. Geitonogamy involves pollination between flowers on the same plant (essentially self-pollination), while xenogamy involves pollination between flowers on different plants (cross pollination).

Response #3.4. The order of the words was changed from ‘xenogamy and geitonogamy’ to ‘geitonogamy and xenogamy’. (Lines 46-47 of the revised manuscript with track changes).

Comment #3.5. For better flow, you may need to move L44 to the end of the paragraph.

Response #3.5. The Line 44 was moved to the end of the paragraph as recommended (see lines 48-49 of the revised manuscript with track changes).

Comment #3.6. Paragraph 1 and 2 are disjointed. I suggest you add a line at the end of paragraph 1 on the importance of the species-mentioning the essential oils as key, to lead the readers to paragraph 2.

Response #3.6. A sentence was added at the end of paragraph 1 (See lines 50-51 of the revised manuscript with track changes)

Comment #3.7. Table 1: The average of Fis is different from what you indicated in L235, please clarify.

Response #3.7. Thank you for the comment. Sorry, the average value of Fis was misreported. The true value was 0.592 and now corrected (See Line 243 of the revised manuscript with track changes). However, the average values of Fis presented in Table 1 are for each studied population. Therefore, to avoid misunderstanding the overall average values of the diversity statistics across all populations, including the Fis were added in Table 1.

Comment #3.8. L243-244 I don’t think the allelic richness among the populations is different. A difference of 0.001 or 0.006 may not be statistically significant. Consider revising.

Response #3.8. True that the difference was not statistically significant. This is now clearly indicated in the sentence that now reads: “Although the mean allelic richness seemed not statistically different among population, the Guinean population presented the highest value, whereas the Sudano-guinean population presented the lowest value for this parameter.” (Lines 250-251 of the revised manuscript with track changes)

Comment #3.9. L251-253 and the table are a bit hard to follow. Consider simplifying them if possible.

Response #3.9. The sentences were improved (Lines 260-264 of the revised manuscript with track changes). The table was improved too (Table 2 of the revised manuscript with track changes).

Comment #3.10. On L350, Fis = 0.59 and is different from the Fis presented in L235 (Fis=0.289), please clarify

Response #3.10. The average value of Fis in (L235) was misreported. The true values was Fis = 0.59 and was corrected accordingly (See Line 243 of the revised manuscript with track changes).

---

## [Editor Report · Decision Letter 3]

20 Aug 2025

SNP markers revealed the genetic diversity and population structure of Mesosphaerum suaveolens (L.) Kuntze Syn. Hyptis suaveolens  (L.) Poit accessions collected in Benin

PONE-D-24-37428R3

Dear Dr. Dossa,

We’re pleased to inform you that your manuscript has been judged scientifically suitable for publication and will be formally accepted for publication once it meets all outstanding technical requirements.

Kind regards,

Mozaniel Santana de Oliveira, Ph.D

Academic Editor

PLOS ONE
---

## [Editor Report · Acceptance letter]

PONE-D-24-37428R3

PLOS ONE

Dear Dr. DOSSA,

I'm pleased to inform you that your manuscript has been deemed suitable for publication in PLOS ONE. Congratulations! Your manuscript is now being handed over to our production team.

Kind regards,

on behalf of

Dr. Mozaniel Santana de Oliveira

Academic Editor

PLOS ONE